# CLIP-LOW INCREASES ENTROPY AND CLIP-HIGH DECREASES ENTROPY IN REINFORCEMENT LEARNING OF LARGE LANGUAGE MODELS

## ABSTRACT

Reinforcement learning with verifiable rewards (RLVR) has recently emerged as the leading approach for enhancing the reasoning capabilities of large language models (LLMs). However, RLVR is prone to entropy collapse, where the LLM quickly converges to a near-deterministic form, hindering exploration and progress during prolonged RL training. In this work, we reveal that the clipping mechanism in PPO and GRPO induces biases on entropy. Through theoretical and empirical analyses, we show that clip-low increases entropy, while clip-high decreases it. Further, under standard clipping parameters, the effect of clip-high dominates, resulting in an overall entropy reduction even when purely random rewards are provided to the RL algorithm. Our findings highlight an overlooked confounding factor in RLVR: independent of the reward signal, the clipping mechanism influences entropy, which in turn affects the reasoning behavior. Furthermore, our analysis demonstrates that clipping can be deliberately used to control entropy. Specifically, with a more aggressive clip-low value, one can increase entropy, promote exploration, and ultimately prevent entropy collapse in RLVR training.

## 1 INTRODUCTION

Reinforcement learning with verifiable rewards (RLVR) has recently emerged as the leading approach for enhancing the reasoning capabilities of large language models (LLMs), especially in the domain of mathematical reasoning (Guo et al., 2025; Lambert et al., 2024; Luong et al., 2024; Yang et al., 2025). However, RLVR is prone to entropy collapse: a phenomenon where the LLM quickly converges to a near-deterministic form, hindering exploration and progress during prolonged RL training (Yu et al., 2025).

Recent studies have reported this effect and continue to debate whether it is an inevitable byproduct of improved performance (Yue et al., 2025; Cui et al., 2025; Wu et al., 2025). A number of works have proposed heuristic interventions to mitigate entropy collapse, such as tuning training hyperparameters (Yu et al., 2025) or explicitly incorporating a KL-divergence loss term (Liu et al., 2025a). Although these approaches can increase policy entropy to some extent, they fall short of providing a mechanistic understanding of why and how entropy evolves during RL training for LLMs.

**Contribution.** In this paper, we elucidate this poorly understood entropy dynamics during RL training of LLMs. First, we theoretically analyze a toy setting where the reward is *random*, i.e., independent of the policy distribution, and we prove that the clipping mechanism used in PPO (Schulman et al., 2017) or GRPO (Shao et al., 2024) induces biases on entropy. Specifically, the lower clip ('clip-low') on negative advantages increases entropy, while the upper clip ('clip-high') on positive advantages decreases entropy. Next, we empirically demonstrate that the theoretical results extend to general RLVR settings for mathematical reasoning tasks. By simply tuning the clipping hyperparameters, we can effectively control the entropy dynamics during RLVR, thereby preventing entropy collapse. Moreover, we show that this entropy-controlled training preserves the base model's exploration capability without compromising its performance, providing a practical tool for stable and prolonged RLVR training.

## 1.1 RELATED WORKS

**Mitigating Entropy collapse in RLVR.** A growing line of work has investigated studied the entropy collapse phenomenon. DAPO (Yu et al., 2025) argued that the clip-high component in PPO (Schulman et al., 2017) and GRPO (Shao et al., 2024) prevents the 'exploration tokens' from being pushed up, accelerating entropy decay. To counter this, they propose 'clip-higher', an asymmetric clipping rule that reduces the clip-high events by setting $\varepsilon_{\text{high}} > \varepsilon_{\text{low}}$. ProRL (Liu et al., 2025a) adopts clip-higher and further emphasizes the use of KL divergence loss for stabilizing entropy; they monitor the training process and manually hard reset the optimization states and reference policy for KL divergence term multiple times to enable prolonged RLVR training. Another popular approach is to use reward shaping to promote exploration (Cheng et al., 2025; Gao et al., 2025a) , which could be understood largely as methods motivated by conventional reinforcement learning algorithms (Haarnoja et al., 2018; Burda et al., 2019). On the other hand, Cui et al. (2025) conducted an extensive search and provided a different viewpoint that the decreasing entropy during training could actually be understood as a tradeoff with peformance, framing entropy collapse as an expected byproduct of training (Deng et al., 2025).

**Exploration of LLMs during RLVR.** There is an active debates about whether RLVR elicits genuinely novel reasoning or merely reweights reasoning paths already latent in the base moel. On one side, recent analyses contend RLVR largely reshapes sampling distributions over pre-existing chain of thought. These works highlight the degradation of the `pass@k` metric during RLVR training(He et al., 2025), and show that post-trained LLMs could underperform the base model when $k$ is large (Yue et al., 2025; Wu et al., 2025). On the other hand, conflicting evidence indicates that RLVR can induce capabilities not present in base models (Wen et al., 2025). For example, carefully reshaping the reward function and deploying an enhaced training schedule has shown to be effective in improving exploration during RLVR (Chen et al., 2025; Song et al., 2025). Notbaly, Liu et al. (2025a) reports cases where RLVR enables solutions to logical tasks that the base model misses even at large $k$. Our findings strengthen this latter perspective: we show that deliberately maintaining higher entropy through controlled clipping could improve `pass@k` without degrading `mean@k`, suggesting that exploration degradation of LLMs is not a inherent limitation of RLVR.

**Random reward for RL.** Counterintuitively, recent studies report that RL can improve LLM benchmark scores even with weak, noisy, or entirely random rewards (Wang et al., 2025; Lv et al., 2025; Zhu et al., 2025). This line of research include methods that utilize *entropy minimization* of the policy model (Zhao et al., 2025; Agarwal et al., 2025; Gao et al., 2025b). The work most closely related to ours is (Shao et al., 2025), where the authors train with purely random rewards and observe gains primarily for models in the `Qwen` family (Yang et al., 2025). We show that, under the hood, entropy minimization is the consistent driver when training with random rewrads, and that this mechanism is appears across a broad set of model families rather than being `Qwen`-specific. This reframes "random-reward improvements" as a predictable consequence of how the clipped RLVR objectives bias policies toward lower-entropy, even when the reward signal provides no information.

## 1.2 NOTATION AND PRELIMINARIES

Consider the setup where given a prompt $x$, an LLM $\pi_\theta$ generates a response $y = (y_1, \ldots, y_T)$ and a reward function $r(y)$ evaluates it. The objective is to maximize expected reward:

$$\underset{\theta}{\text{maximize}} \quad \mathcal{J}(\theta) := \underset{\substack{x \sim \mathcal{D} \\ y \sim \pi_\theta(\cdot|x)}}{\mathbb{E}} [r(y)], \tag{1}$$

where $\mathcal{D}$ denotes the training distribution of prompts.

We formulate this optimization problem into an RL problem. Specifically, consider the MDP with a discrete state space $\mathcal{S}$ and a finite action space $\mathcal{A}$ is the finite action space. The state is defined as $s_t = (x, y_1, \ldots, y_{t-1})$ and action $a_t$ is the next token to generate, and the transition dynamics is a deterministic one in which the generated token is appended to the state. Finally, the language model $\pi_\theta$ is regarded as the policy, and we refer to this as the reinforcement learning of large language models (RL-LLM) setup.

Given a policy (language model) $\pi$, we define its state visitation measure as

$$d^\pi(s) = \sum_{t=0}^{\infty} \mathbb{P}(s_t = s) = \mathbb{E}\Big[\sum_{t=0}^{T} \mathbf{1}_{s_t = s}\Big],$$

where the probability and expectation is with respect to $s_0 = x \sim \mathcal{D}$ and $a_t \sim \pi(\cdot \,|\, s_t)$ for $t = 0, 1, \ldots$.

**REINFORCE.** The classical REINFORCE policy gradient estimator (Williams, 1992) is given by

$$\nabla_\theta \mathcal{J}(\theta) = \mathop{\mathbb{E}}_{\substack{x \sim \mathcal{D} \\ y \sim \pi_\theta(\cdot|x)}} \left[ \sum_{t=1}^{T} \nabla_\theta \log \pi_\theta(y_t \,|\, y_{<t}, x) \, A_t \right], \tag{2}$$

where $y_{<t} := (y_1, \ldots, y_{t-1})$ and $A_t$ is an advantage estimate derived from the trajectory-level rewards, such as $A_t = r(y_T)$.

Although it is possible to perform stochastic gradient descent (ascent) using the stochastic gradients from Equation 2 (and doing so would avoid the clipping bias that we identify in this work), such an approach is typically less sample-efficient and less stable. Therefore, methods such as PPO (Schulman et al., 2017) and GRPO (Shao et al., 2024) are preferred in the RL-LLM setting.

**Group Relative Policy Optimization (GRPO).** GRPO (Shao et al., 2024) is a variant of proximal policy optimization (PPO) (Schulman et al., 2017) adapted for trajectory-level rewards. Given a current policy parameter $\theta_{\text{old}}$, the algorithm samples a prompt $x \sim \mathcal{D}$ and $K$ responses $y^{(1)}, \ldots, y^{(K)} \sim \pi_{\theta_{\text{old}}}(\cdot \,|\, x)$. Then, the parameter update to $\theta$ is obtained by performing stochastic gradient steps to solve the subproblem

$$\underset{\theta}{\text{minimize}} \quad \sum_{i=1}^{K} \frac{1}{T^{(i)}} \sum_{t=1}^{T^{(i)}} \min\Big( r_t^{(i)}(\theta) \, A_t^{(i)}, \; \text{clip}\big(r_t^{(i)}(\theta), \, 1 - \varepsilon_{\text{low}}, \, 1 + \varepsilon_{\text{high}}\big) A_t^{(i)} \Big)$$

with

$$r_t^{(i)}(\theta) = \frac{\pi_\theta\big(y_t^{(i)} \,\big|\, y_{<t}^{(i)}, x\big)}{\pi_{\theta_{\text{old}}}\big(y_t^{(i)} \,\big|\, y_{<t}^{(i)}, x\big)}, \qquad A_t^{(i)} = r(y^{(i)}) - \text{mean}\big(r(y^{(1)}), \ldots, r(y^{(K)})\big)$$

for $t = 1, \ldots, T^{(i)}$ and $i = 1, \ldots, K$.

The clipping mechanism, whose strength is controlled by the hyperparameters $\varepsilon_{\text{low}}$ and $\varepsilon_{\text{high}}$, originates from trust-region policy optimization (TRPO) (Schulman et al., 2015). Its purpose is to prevent the optimization for the subproblem from deviating too far from the reference policy $\pi_{\theta_{\text{old}}}$ that generated the responses. Concretely, the importance sampling ratio $r_t^{(i)}(\theta)$ is clipped to lie within the range $[1 - \varepsilon_{\text{low}}, 1 + \varepsilon_{\text{high}}]$ depending on the sign of $A_t^{(i)}$. The main thesis of this paper is that the two clipping mechanisms induce biases on entropy.

To be precise, the version of GRPO we present here is more closely aligned with the variant called DAPO (Yu et al., 2025). While the original GRPO formulation (Shao et al., 2024) normalizes $A_t^{(i)}$ by the standard deviation of the rewards, we follow the prescription of Dr. GRPO (Liu et al., 2025b) and omit this normalization. In addition, whereas the original PPO and GRPO employ a symmetric clipping parameter with $\varepsilon_{\text{low}} = \varepsilon_{\text{high}}$, DAPO introduces asymmetric clipping with $\varepsilon_{\text{low}} < \varepsilon_{\text{high}}$.

**Policy entropy.** For any state $s_t$, the token-level (state-conditional) Shannon entropy of the policy $\pi_\theta$ is defined as

$$\mathcal{H}(\pi_\theta \,|\, s_t) = - \sum_{a \in \mathcal{A}} \pi_\theta(a \,|\, s_t) \log \pi_\theta(a \,|\, s_t), \tag{3}$$

where $\mathcal{A}$ (note, $|\mathcal{A}| < \infty$) is the LLM vocabulary. In practice, we report the average token entropy over responses, evaluated over states encountered under the old policy distribution $\pi_\theta$. For a minibatch of size $N$, we estimate the entropy with the following formula

$$\hat{\mathcal{H}}(\pi_\theta) = -\frac{1}{N} \sum_{i=1}^{N} \left[ \frac{1}{T^{(i)}} \sum_{t=1}^{T^{(i)}} \mathcal{H}(\pi_\theta \,|\, s_t^{(i)}) \right]. \tag{4}$$

## 2 THEORETICAL ANALYSIS OF CLIPPING WITH RANDOM REWARDS

Following the formulation of Shao et al. (2025), we consider the setting of *random rewards* for the sake of theoretical analysis and scientific inquiry. Specifically, the random rewards are assumed to be statistically independent of both the prompt and the response generated by the LLM, and to have a symmetric distribution (e.g., a reward that takes values $0$ and $1$ with equal probability is symmetric about $1/2$), which in turn leads to GRPO-style advantage estimates having a zero-mean, symmetric distribution.

By construction, such random rewards and the corresponding advantage estimates computed from them contain no learning signal. Indeed, the associated REINFORCE-type policy gradient estimator has zero expectation:

$$
\mathop{\mathbb{E}}_{\substack{x\sim\mathcal{D}\\ y_t\sim\pi_\theta(\cdot|y_{<t},x)\\ A}}\left[\sum_{t=1}^T \nabla_\theta \log \pi_\theta(y_t|y_{<t})A\right] = \mathop{\mathbb{E}}_{\substack{x\sim\mathcal{D}\\ y_t\sim\pi_\theta(\cdot|y_{<t},x)}}\left[\sum_{t=1}^T \nabla_\theta \log \pi_\theta(y_t|y_{<t})\right]\mathbb{E}[A] = 0.
$$

However, GRPO and its variants crucially employ a clipping mechanism, and in this section, we show that this clipping mechanism induces biases on entropy.

### 2.1 SETUP FOR THE THEORETICAL ANALYSIS

Consider the objective function of the GRPO subproblem:

$$
\mathcal{J}(\pi;\pi_{\text{old}}) = \mathop{\mathbb{E}}_{\substack{x\sim\mathcal{D}\\ y\sim\pi_{\text{old}}(\cdot|x)\\ A}}\left[\frac{1}{T}\sum_{t=1}^T \min\left(\frac{\pi(y_t\,|\,y_{<t},x)}{\pi_{\text{old}}(y_t\,|\,y_{<t},x)}\,A,\ \text{clip}\left(\frac{\pi(y_t\,|\,y_{<t},x)}{\pi_{\text{old}}(y_t\,|\,y_{<t},x)},\,1-\varepsilon_{\text{low}},\,1+\varepsilon_{\text{high}}\right)A\right)\right].
$$

We assume the advantage $A$ is independent of of $x$ and $y$ and satisfies

$$
\mathbb{E}[A] = 0, \quad \mathbb{P}(A>0) = \mathbb{P}(A<0) = \nu, \quad \mathbb{E}[A\,|\,A>0] = \mu.
$$

The actual GRPO algorithm performs a limited number of optimization steps on the objective $\mathcal{J}$, typically using AdamW, which is difficult to model and analyze directly. For the sake of analytical tractability, we assume the use of full batch gradients and consider two simplified formulations: the policy gradient and natural policy gradient algorithms applied to $\mathcal{J}$. Namely, the first algorithm is the policy gradient algorithm

$$
\theta_{k+1} = \theta_k + \eta\nabla_\theta\mathcal{J}(\pi_{\theta_k};\pi_{\text{old}}), \tag{5}
$$

where $\pi_{\text{old}}$ is an older version of $\pi_{\theta_k}$ that is updated by the outer loop of GRPO and $\pi_\theta$ is parameterized as a tabular softmax policy

$$
\pi_\theta(a|s) = \frac{\exp(\theta_{s,a})}{\sum_{a'\in\mathcal{A}}\exp(\theta_{s,a'})} \qquad \text{for } s\in\mathcal{S},\,a\in\mathcal{A}
$$

with state space $\mathcal{S}$, finite action space $\mathcal{A}$, and trainable parameter $\theta\in\mathbb{R}^{|\mathcal{S}|\times|\mathcal{A}|}$. The second algorithm is the *natural* policy gradient algorithm (Kakade, 2001)

$$
\pi_{k+1} \propto \pi_k \circ \exp\left(\eta\nabla_\pi\mathcal{J}(\pi_k;\pi_{\text{old}})\right), \tag{6}
$$

where again $\pi_{\text{old}}$ is an older version of $\pi_k$ that is updated by the outer loop of GRPO and $\circ$ denotes element-wise multiplication. As we will see, our analysis of the two algorithms yields results that differ slightly but are qualitatively aligned. Since the two algorithms are considered models of the true GRPO update, this consistency lends further credibility to the qualitative conclusions drawn from our analysis.

Now, define the following probabilistic events

$$
X_k(s) = \left\{\text{event such that } \frac{\pi_k(a|s)}{\pi_{\text{old}}(a|s)} < 1-\varepsilon_{\text{low}}\right\} = \left\{\text{event such that clip-low happens}\right\}
$$

$$
Y_k(s) = \left\{\text{event such that } \frac{\pi_k(a|s)}{\pi_{\text{old}}(a|s)} > 1+\varepsilon_{\text{high}}\right\} = \left\{\text{event such that clip-high happens}\right\}.
$$

Whether events $X_k(s)$ and $Y_k(s)$ hold is determined by the action $a\sim\pi_{\text{old}}(\cdot\,|\,s)$.

## 2.2 FIRST-ORDER ANALYSIS OF ENTROPY CHANGE

We first present our analysis of the entropy change of the policy gradient algorithm.

**Theorem 1.** *Consider the setup described in Section 2.1 and the policy gradient algorithm given by Equation 5. Then, the change in entropy at state $s$ admits the first-order approximation*

$$\mathcal{H}(\theta_{k+1}\,|\,s) - \mathcal{H}(\theta_k\,|\,s) = \mu\nu\eta\; d^{\pi_{\text{old}}}\big(\underbrace{p_k(\mathbb{E}[Q] - \mathbb{E}[Q\,|\,X_k])}_{\text{clip-low contribution}} - \underbrace{q_k(\mathbb{E}[Q] - \mathbb{E}[Q\,|\,Y_k])}_{\text{clip-high contribution}}\big) + \mathcal{O}(\eta^2)$$

*where $Q = \pi_k(a\,|\,s)(\log \pi_k(a\,|\,s) + \mathcal{H}(\theta^k\,|\,s))$, $p_k = \mathbb{P}(X_k)$, $q_k = \mathbb{P}(Y_k)$, $d^{\pi_{\text{old}}}$ is the state visitation measure, and the expectation $\mathbb{E}$ is taken with respect to $a \sim \pi_k(\cdot\,|\,s)$. To clarify, all the terms on the right-hand side depend on $s$, and it would be more precise to write them as $d^{\pi_{\text{old}}}(s)$, $Q(s)$, $X_k(s)$, $Y_k(s)$, $p_k(s)$, and $q_k(s)$. However, we suppress the dependence on $s$ for notational simplicity.*

We defer the proof to Appendix A.

Theorem 1 separates the contributions of clip-low and clip-high. Decreasing $\varepsilon_{low}$ leads to a larger $p_k = \mathbb{P}(X_k)$, thereby amplifying the clip-low term, and vice-versa for clip-high. Moreover, if either clip-low or clip-high is turned off, $p_k = 0$ or $q_k = 0$, and only the other term remains.

If the following condition holds:

$$\mathbb{E}[Q] - \mathbb{E}[Q\,|\,X_k] \geq 0 \quad \text{and} \quad \mathbb{E}[Q] - \mathbb{E}[Q\,|\,Y_k] \geq 0, \tag{7}$$

then the claim that clip-low increases entropy and clip-high decreases entropy is substantiated. Inequalities 7, however, are not guaranteed to hold universally, and counterexamples can be constructed where the condition fails. Nevertheless, we empirically observe that Inequalities 7 are typically satisfied in practice. In particular, Figure 1 shows that empirical estimates consistently meet these conditions.

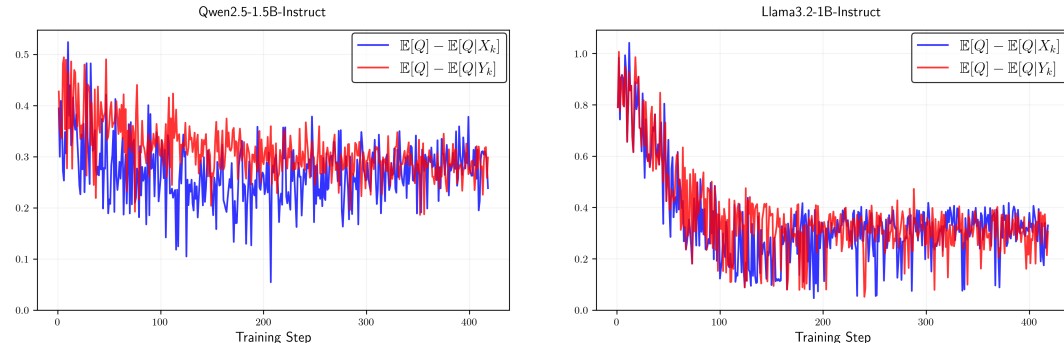

Figure 1: Empirical estimates of $\mathbb{E}[Q] - \mathbb{E}[Q\,|\,X_k]$ and $\mathbb{E}[Q] - \mathbb{E}[Q\,|\,Y_k]$ throughout RL training with random rewards for **(Left)** `Qwen2.5-1.5B-Instruct` and **(Right)** `Llama3.2-1B-Instruct`. We observe that the values are always positive.

Next, we present our analysis of the entropy change of the *natural* policy gradient algorithm.

**Theorem 2.** *Consider the setup described in Section 2.1 and the natural policy gradient algorithm given by Equation 6. Then, the change in entropy at state $s$ admits the first-order approximation*

$$\mathcal{H}(\pi_{k+1}\,|\,s) - \mathcal{H}(\pi_k\,|\,s)$$
$$= \mu\nu\eta\; d^{\pi_{\text{old}}}\big(\underbrace{p_k(\mathbb{E}[-\log \pi_k\,|\,X_k] - \mathcal{H}(\pi_k\,|\,s))}_{\text{clip-low contribution}} - \underbrace{q_k(\mathbb{E}[-\log \pi\,|\,Y_k] - \mathcal{H}(\pi_k\,|\,s))}_{\text{clip-high contribution}}\big) + \mathcal{O}(\eta^2),$$

*where $p_k = \mathbb{P}(X_k)$, $q_k = \mathbb{P}(Y_k)$, $d^{\pi_{\text{old}}}$ is the state visitation measure, and the expectation $\mathbb{E}$ is taken with respect to $a \sim \pi_k(\cdot\,|\,s)$. To clarify, all the terms on the right-hand side depend on $s$, and it would be more precise to write them as $d^{\pi_{\text{old}}}(s)$, $X_k(s)$, $Y_k(s)$, $p_k(s)$, and $q_k(s)$. However, we suppress the dependence on $s$ for notational simplicity.*

We defer the proof to Appendix B.

Theorem 2 again separates the contributions of clip-low and clip-high. If the following condition holds:

$$\mathbb{E}[-\log \pi_k \mid X_k] - \mathcal{H}(\pi_k \mid s) \geq 0 \quad \text{and} \quad \mathbb{E}[-\log \pi \mid Y_k] - \mathcal{H}(\pi_k \mid s) \geq 0, \tag{8}$$

then the claim that clip-low increases entropy and clip-high decreases entropy is substantiated. Again, we empirically observe that Inequalities 8 are typically satisfied in practice. In particular, Figure 2 shows that empirical estimates consistently meet these conditions.

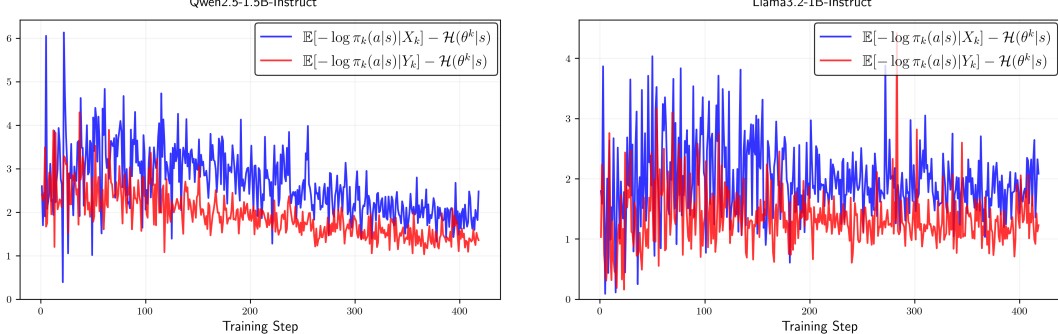

Figure 2: Estimated values of equation 8 throughout RL training with random rewards averaged over 3 runs. **(left)** `Qwen2.5-1.5B-Instruct` and **(right)** `Llama3.2-1.5B-Instruct`. We observe that the values are always positive.

### 2.3 EMPIRICAL VALIDATION

In this section, we present an empirical validation of our theory.

**Setting.** We use the `verl` framework (Sheng et al., 2025) for all experiments. The models are trained with the `GSM8K` dataset (Cobbe et al., 2021) but the rewards are randomly drawn from a Bernoulli distribution with $0.5$ probability. We use the GRPO algorithm and, following Dr. GRPO (Liu et al., 2025b), we do not normalize rewards by the standard deviation in the advantage calculation. We use the `Qwen2.5-3B-Instruct` (Yang et al., 2024) and `Llama3-8B-Instruct` models as our base models. We use a GRPO batch size of $512$, and an optimizer batch size of $256$. Neither the KL divergence loss nor an explicit entropy loss is applied. For each rollout, we generate 8 prompts with temperature $T = 1$. We use the `AdamW` optimizer with a constant learning rate of $5 \cdot 10^{-7}$. During validation rollout, we use temperature $T = 0.6$. We defer further implementation details to Appendix C.1.

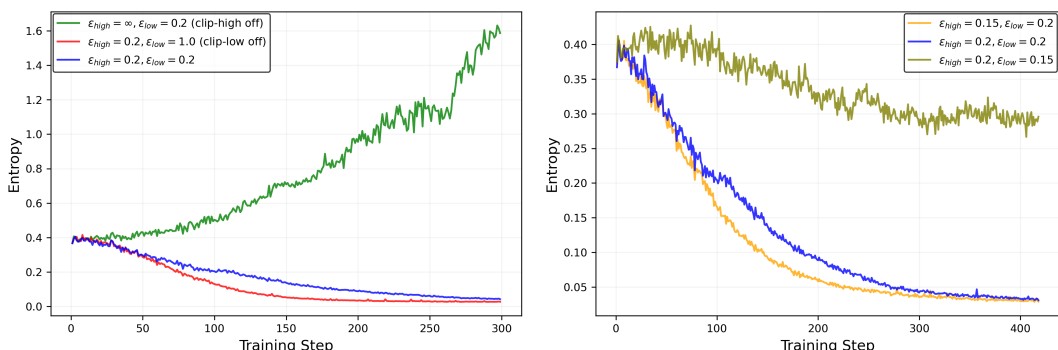

Figure 3: Change of policy entropy during RL training the `Qwen2.5-1.5B-Instruct` model with random rewards with different clipping settings. We observe that both clip-high and clip-low influence the entropy, consistent with our theoretical predictions.

**Results.** The experimental results are consistent with our theoretical predictions. Figure 3 shows that decreasing/increasing $\varepsilon_{\text{low}}$ (making clip-low stronger/weaker) increases/decreases entropy, and decreasing/increasing $\varepsilon_{\text{high}}$ (making clip-high stronger/weaker) decreases/increases entropy.

Moreover, we find that with symmetric clipping parameters ($\varepsilon_{\text{low}} = \varepsilon_{\text{high}} = 0.2$), the effect of clip-high dominates that of clip-low, leading to a reduction in entropy. However, by appropriately decreasing $\varepsilon_{\text{low}}$ (making clip-low stronger), we can counterbalance the competing effects and maintain the entropy level.

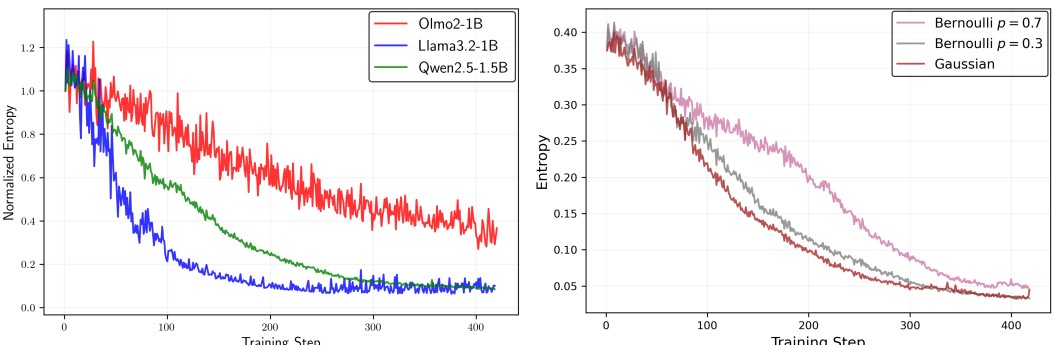

Figure 4: **(Left)** Entropy change of different base models when trained with random rewards under symmetric clipping $\varepsilon_{\text{low}} = \varepsilon_{\text{high}}$. **(Right)** Entropy change of `Qwen2.5-1.5B-Instruct` model with random rewards sampled from various probability distributions. Details of the experiments are provided in Appendix C.2.

**Noisy and spurious rewards reduce entropy.** Prior work has investigated whether RLVR can enhance LLM reasoning even in the presence of noisy rewards (Wang et al., 2025; Lv et al., 2025) or random (spurious) rewards (Shao et al., 2025). In particular, Shao et al. (2025) find that GRPO-based training with clipping yields clear improvements for `Qwen`-based models, but little to no benefit for `Llama`- or `Olmo`-based models. By contrast, Figure 4 shows that training with random rewards consistently reduces policy entropy across `Qwen`, `Llama`, and `Olmo`. This pattern suggests that the primary effect may be entropy minimization, which in turn influences reasoning behavior as recently suggested in (Agarwal et al., 2025; Gao et al., 2025b).

## 3 EMPIRICAL ANALYSIS OF CLIPPING WITH RLVR

In this section, we extend the theoretical insights from the random reward setting of Section 2 to the general (true reward) RLVR setting through empirical analysis. Our results demonstrate that the clipping parameters, $\varepsilon_{\text{high}}$ and $\varepsilon_{\text{low}}$, provide effective control over policy entropy in RLVR for mathematical reasoning tasks. Moreover, such entropy control improves the exploration (as measured by `pass@8`) while preserving reasoning performance (as measured by `mean@8`). Specifically, the

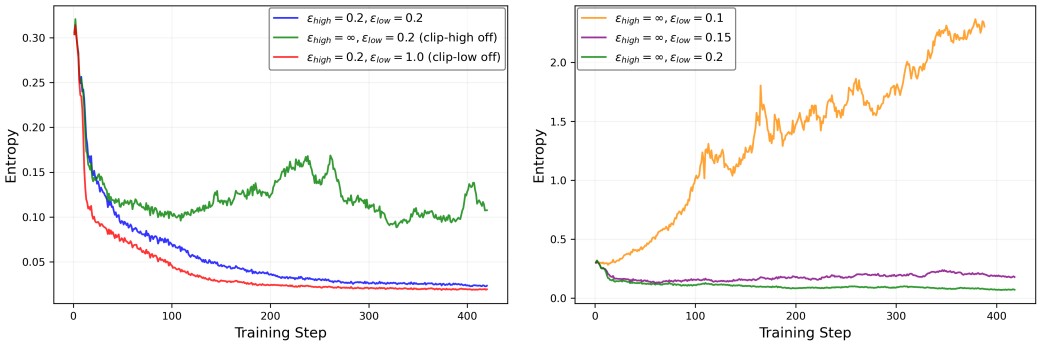

Figure 5: Entropy change during (true reward) RLVR with `GSM8K` and `Qwen2.5-3B-Instruct`. **(left)** Ablating the clipping mechanisms. **(right)** Controlling entropy without clip-high. The clip-low value $\varepsilon_{\text{low}} = 0.15$ balances entropy, preventing entropy collapse and entropy explosion.

`pass@8` metric measures whether at least one of the 8 sampled responses yields the correct solution (Chen et al., 2021), while `mean@8` reflects the average single-response accuracy (`pass@1`) across those 8 responses.

### 3.1 EXPERIMENTAL SETUP

Again, we use the `verl` framework (Sheng et al., 2025) for the RL training and `GSM8K` (Cobbe et al., 2021) and the `DAPO-Math-17k` (Yu et al., 2025) for the mathematical reasoning training data. For `GSM8K`, we use `Qwen2.5-3B-Instruct` and `Llama3-8B-Instruct` as base models, and for the `DAPO-Math-17k` dataset, we use `Qwen2.5-7B-Instruct` as the base model. We use the same configurations for the GRPO algorithm as in our random reward experiments of Section 2.3. Refer to Appendix C.1 for further training details.

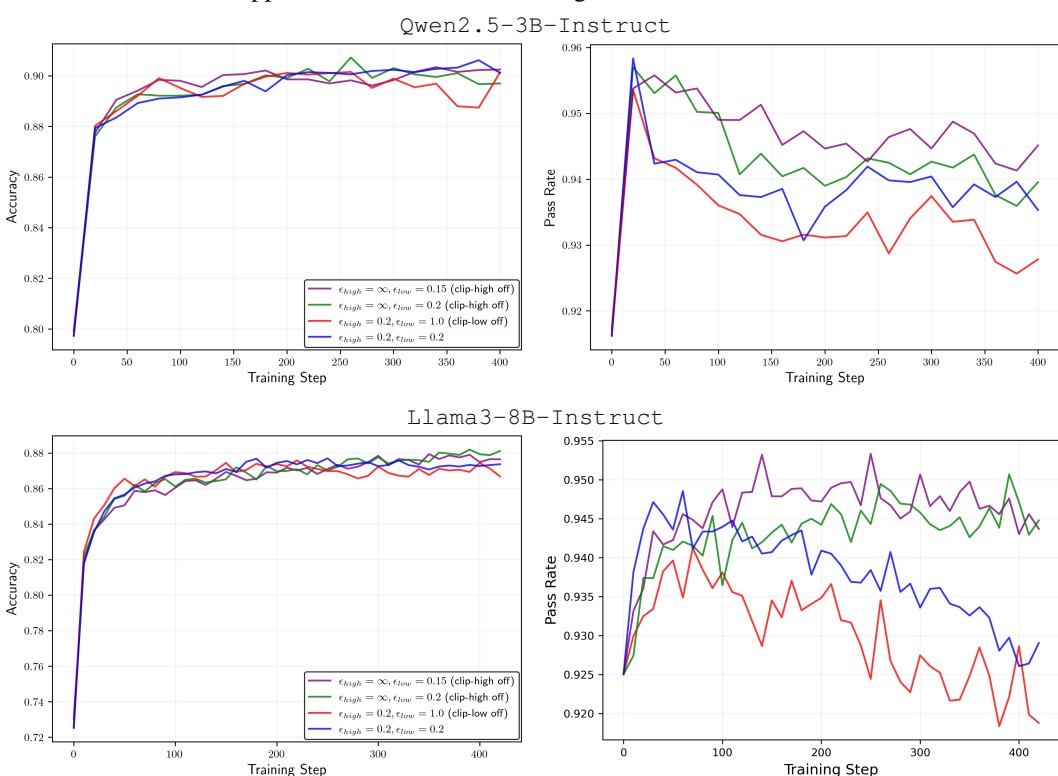

Figure 6: Performance of LLM during RLVR training with `GSM8K` dataset measured by the **(left)** `mean@8` metric and **(right)** `pass@8` metric for **(up)** `Qwen2.5-3B-Instruct` model and **(down)** `Llama3-8B-Instruct` model. While all settings configurations show comparable `mean@8` performance, training setups with high entropy show higher `pass@8` performance, implying enhanced exploration.

### 3.2 EXPERIMENTS: MATH REASONING TASKS

**Clip-high decreases entropy and clip-low increases entropy.** We begin with an ablation study of the clipping mechanisms. Specifically, we disable the clip-low mechanism (by setting $\varepsilon_{\text{low}} = 1.0$) and the clip-high mechanism (by setting $\varepsilon_{\text{high}} = \infty$). As shown in Figure 5 (left), removing clip-high increases entropy, while removing clip-low decreases it, in qualitative agreement with the theoretical analysis for the random reward setting in Section 2.

**Entropy control via Clip-Lower** Unlike the random reward setting, RLVR training with true rewards has an entropy-reduction effect, which can be attributed to RLVR's suppression of incorrect reasoning paths. For example, while the configuration $\varepsilon_{\text{high}} = \infty$ (clip-high off) and $\varepsilon_{\text{low}} = 0.2$ increased entropy in the random reward setting (Figure 3, left), the same configuration leads to reduced entropy in the true reward RLVR setup (Figure 5).

To counteract RLVR's natural entropy reduction, turn off clip-high ($\varepsilon_{\text{high}} = \infty$) and adjust the clip-low parameter $\varepsilon_{\text{low}}$ to a smaller value. As shown in Figure 5 (right), decreasing $\varepsilon_{\text{low}}$ increases entropy during training—sometimes to the extreme of entropy explosion. For this particular setup, we find that the configuration ($\varepsilon_{\text{high}} = \infty, \varepsilon_{\text{low}} = 0.15$) achieves a balance, preventing both entropy collapse and entropy explosion.

**Entropy control leads to improved exploration.** While RLVR enhances the reasoning performance of LLMs, prior work (Yue et al., 2025; Song et al., 2025) has shown that it also narrows the range of reasoning trajectories the model can explore, also referred to as the *reasoning boundary*. Consistent with this, Figures 6 and 7 shows that training with the standard symmetric clipping parameters ($\varepsilon_{low} = \varepsilon_{high} = 0.2$) causes the `pass@8` metric to decline over the course of training.

However, when entropy is controlled through clipping (entropy is shown in Figure 5), the `pass@8` metric is preserved without sacrificing the `mean@8` performance as shown in Figure 6. Moreover, Figure 7 shows that the clipping mechanisms can be tuned to simultaneously improve the `mean@8` and `pass@8` performances. These results demonstrate that entropy collapse can be avoided through appropriate clipping parameter choices, even without a KL penalty. Moreover, they confirm that this entropy control does genuinely correspond to exploration.

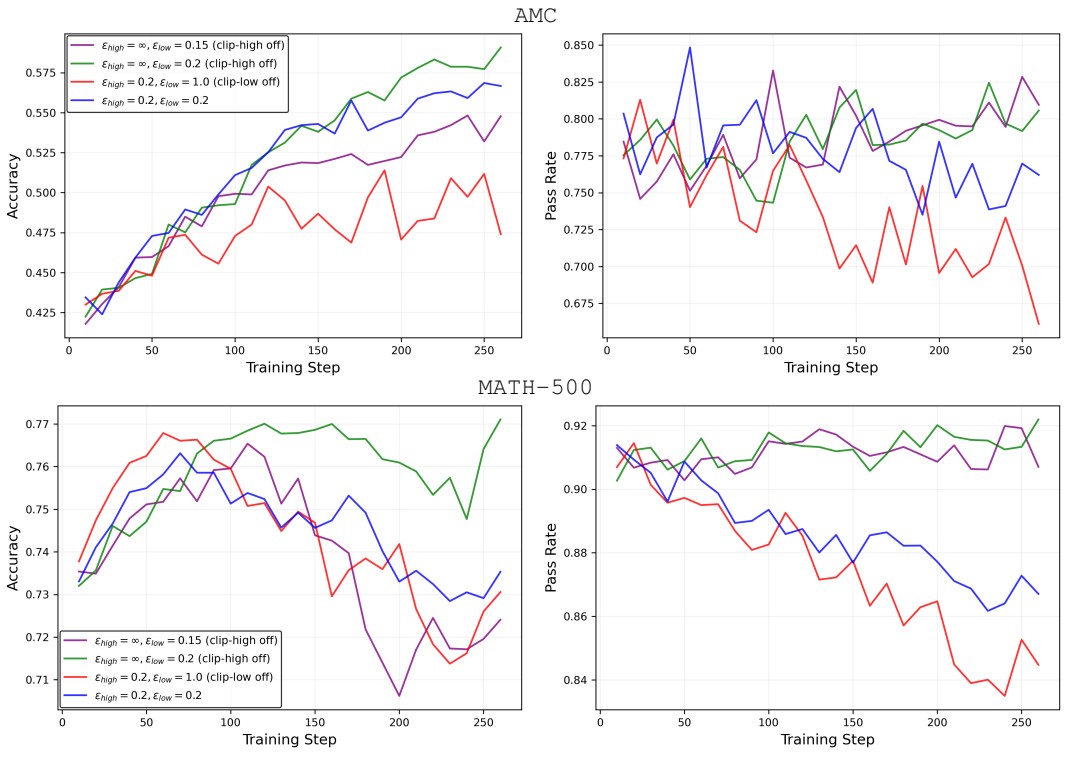

Figure 7: Performance measured by the `mean@32` metric **(left)** and `pass@32` metric **(right)** metric during RLVR for the `Qwen2.5-7B-Instruct` model trained with `DAPO-Math-17k` dataset, evaluated on the `AMC` and `MATH-500` datasets.

## 4 CONCLUSION

In this work, we reveal that the clipping mechanism in PPO and GRPO induces biases on entropy, thereby highlighting an overlooked confounding factor in RLVR. Furthermore, we demonstrate that the entropy can be controlled by appropriately setting the clip-low and clip-high values.

Our findings open up several promising avenues for future research. One is to expand the theory by relaxing the assumptions and filling in the theoretical gaps. Another is to empirically investigate how clipping can be utilized to maximize performance. Notably, such performance optimization may correlate with, but is not equivalent to, simply maintaining an appropriate level of entropy.

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

## A   ANALYSIS OF POLICY GRADIENT: PROOF OF THEOREM 1

Here we present the proof for Theorem 1.

*Proof.* We first analyze the first-order Taylor expansion of entropy relative to logit change ($\Delta\theta_{s,a} = \theta_{s,a}^{k+1} - \theta_{s,a}^{k}$). This first step is closely inspired by Liu (2025). We can Taylor expand the entropy with respect to $\Delta\theta$:

$$\mathcal{H}(\theta^{k+1}|s) = \mathcal{H}(\theta^k|s) + \left\langle \nabla_\theta \mathcal{H}(\theta^k|s), \ \Delta\theta \right\rangle + \mathcal{O}((\Delta\theta)^2)$$

The gradient of policy entropy is

$$\begin{aligned}
\nabla_\theta \mathcal{H}(\theta|s) &= \nabla_\theta \left( -\mathbb{E}_{a\sim\pi_\theta(\cdot|s)}[\log \pi_\theta(a|s)] \right) \\
&= -\mathbb{E}_{a\sim\pi_\theta(\cdot|s)}\left[ \nabla_\theta \log \pi_\theta(a|s) + \log \pi_\theta(a|s) \nabla_\theta \log \pi_\theta(a|s) \right] \\
&= -\mathbb{E}_{a\sim\pi_\theta(\cdot|s)}\left[ \log \pi_\theta(a|s) \nabla_\theta \log \pi_\theta(a|s) \right].
\end{aligned}$$

Therefore, we have

$$\begin{aligned}
\left\langle \nabla_\theta \mathcal{H}(\theta^k|s), \theta^{k+1} - \theta^k \right\rangle &= -\left\langle \mathbb{E}_{a\sim\pi_k(\cdot|s)}\left[ \log \pi_k(a|s) \nabla_\theta \log \pi_k(a|s) \right], \ \theta^{k+1} - \theta^k \right\rangle \\
&= -\mathbb{E}_{a\sim\pi_k(\cdot|s)}\left[ \log \pi_k(a|s) \left\langle \nabla_\theta \log \pi_k(a|s), \ \theta^{k+1} - \theta^k \right\rangle \right] \\
&= -\mathbb{E}_{a\sim\pi_k(\cdot|s)}\left[ \log \pi_k(a|s) \sum_{s'\in\mathcal{S},a'\in\mathcal{A}} \frac{\partial \log \pi_k(a|s)}{\partial \theta_{s',a'}} \cdot \left( \theta_{s',a'}^{k+1} - \theta_{s',a'}^{k} \right) \right] \\
&= -\sum_{s'\in\mathcal{S},a'\in\mathcal{A}} \mathbb{E}_{a\sim\pi_k(\cdot|s)}\left[ \log \pi_k(a|s) \cdot \frac{\partial \log \pi_k(a|s)}{\partial \theta_{s',a'}} \right] \cdot \left( \theta_{s',a'}^{k+1} - \theta_{s',a'}^{k} \right) \\
&= -\sum_{s'\in\mathcal{S},a'\in\mathcal{A}} \left( \theta_{s',a'}^{k+1} - \theta_{s',a'}^{k} \right) \cdot \mathbb{E}_{a\sim\pi_k(\cdot|s)}\left[ \log \pi_k(a|s) \cdot \frac{\partial \log \pi_\theta(a|s)}{\partial \theta_{s',a'}} \right] \\
&\overset{(\star)}{=} -\sum_{s'\in\mathcal{S},a'\in\mathcal{A}} \left( \theta_{s',a'}^{k+1} - \theta_{s',a'}^{k} \right) \cdot \mathbf{1}_{\{s=s'\}} \cdot \pi_k(a'|s) \left( \log \pi_k(a'|s) - \mathbb{E}_{a\sim\pi_k(\cdot|s)}[\log \pi_k(a|s)] \right)
\end{aligned}$$

where the final equation holds from the derivation below.

$$\begin{aligned}
\mathbb{E}_{a\sim\pi_k(\cdot|s)}\left[ \log \pi_k(a|s) \cdot \frac{\partial \log \pi_\theta(a|s)}{\partial \theta_{s',a'}} \right] &= \mathbb{E}_{a\sim\pi_k(\cdot|s)}\left[ \log \pi_k(a|s) \cdot \frac{\partial}{\partial \theta_{s',a'}} \left( \theta_{s,a} - \log \left( \sum_{a\in\mathcal{A}} \exp\{\theta_{s,a}\} \right) \right) \right] \\
&= \mathbb{E}_{a\sim\pi_k(\cdot|s)}\left[ \log \pi_k(a|s) \cdot \mathbf{1}_{\{s=s'\}} \cdot \left( \mathbf{1}_{\{a=a'\}} - \pi_k(a'|s) \right) \right] \\
&= \mathbf{1}_{\{s=s'\}} \cdot \mathbb{E}_{a\sim\pi_k(\cdot|s)}\left[ \log \pi_k(a|s) \cdot \left( \mathbf{1}_{\{a=a'\}} - \pi_k(a'|s) \right) \right] \\
&= \mathbf{1}_{\{s=s'\}} \cdot \left[ \pi_k(a'|s) \log \pi_k(a'|s) - \pi_k(a'|s) \cdot \mathbb{E}_{a\sim\pi_k(\cdot|s)}[\log \pi_k(a|s)] \right].
\end{aligned}$$

Hence, we obtain the first-order Taylor expansion of policy entropy:

$$\mathcal{H}(\theta^{k+1}|s) - \mathcal{H}(\theta^k|s) = -\mathbb{E}_{a\sim\pi_k(\cdot|s)}\left[ \left( \theta_{s,a}^{k+1} - \theta_{s,a}^{k} \right) \left( \log \pi_k(a|s) + \mathcal{H}(\theta^k|s) \right) \right] + \mathcal{O}((\Delta\theta)^2). \quad (9)$$

For our next step (and this is where the technical novelty of our analysis begins), we express the logit change $\Delta\theta$ in terms of clipping events. Consider the clipped surrogate objective

$$\mathcal{J}(\theta) = \mathbb{E}_{x\sim\mathcal{D}, y\sim\pi_{old}(\cdot|x), A}\left[ \frac{1}{T} \sum_{t=0}^{T} C_\varepsilon(r_t, A_t) \right]$$

where $r_t = \frac{\pi_\theta(y_t|y_{<t},x)}{\pi_{old}(y_t|y_{<t},x)}$. Now we compute the partial derivative of $\mathcal{J}(\theta)$ over each $\theta_{s,a}$. Here since $\pi_\theta(a|s)$ is a function of $\theta_{\cdot,s}$, $C_\varepsilon(r_t, A_t)$ is a constant with respect to $\theta_{s,a}$ unless $s = (y_{<t}, x)$.

Therefore

$$\frac{\partial}{\partial \theta_{s,a}} \mathcal{J}(\theta) = \mathbb{E}_{x \sim \mathcal{D}, \tau \sim \pi_{old}(\cdot|x), A} \left[ \frac{1}{T} \frac{\partial}{\partial \theta_{s,a}} \sum_{t=0}^{T} C_\varepsilon(r_t, A_t) \right]$$

$$= \mathbb{E}_{x \sim \mathcal{D}, \tau \sim \pi_{old}(\cdot|x), A} \left[ \frac{1}{T} \sum_{t=0}^{T} \frac{\partial}{\partial \theta_{s,a}} \mathbf{1}_{\{(y_{<t},x)=s\}} C_\varepsilon(r_t, A_t) \right]$$

$$= \mathbb{E}_{x \sim \mathcal{D}, y_t \sim \pi_{old}(\cdot|y_{<t},x), A_t} \left[ \mathbf{1}_{\{(y_{<t},x)=s\}} \frac{\partial}{\partial \theta_{s,a}} C_\varepsilon(r_t, A_t) \right]$$

$$= d^{\pi_{old}}(s) \times \mathbb{E}_{a' \sim \pi_{old}(\cdot|s), A} \left[ \frac{\partial}{\partial \theta_{s,a}} C_\varepsilon(r(s, a'), A) \right]$$

where $d^{\pi_{old}}(s)$ is the state-visiting probability under the policy $\pi_{old}$. Thus we can write

$$\frac{1}{d^{\pi_{old}}(s)} \frac{\partial}{\partial \theta_{s,a}^k} \mathcal{J}(\theta) = \mathbb{E}_{x \sim \mathcal{D}, a' \sim \pi_{old}(\cdot|s), A} \left[ \frac{\partial}{\partial \theta_{s,a}} C_\varepsilon(r(s, a'), A) \right]$$

$$= \mathbb{P}(A > 0) \mathbb{E}_{a' \sim \pi_{old}(\cdot|s), A} \left[ \frac{\partial}{\partial \theta_{s,a}} C_\varepsilon(r(s, a'), A) \mid A > 0 \right] + \mathbb{P}(A < 0) \mathbb{E}_{a' \sim \pi_{old}(\cdot|s), A} \left[ \frac{\partial}{\partial \theta_{s,a}} C_\varepsilon(r(s, a'), A) \mid A < 0 \right]$$

$$= \mathbb{P}(A > 0, 1 - \varepsilon < r(s, a') < 1 + \varepsilon) \mathbb{E}_{a' \sim \pi_{old}(\cdot|s), A} \left[ \frac{\partial}{\partial \theta_{s,a}} C_\varepsilon(r(s, a'), A) \mid A > 0, 1 - \varepsilon < r(s, a') < 1 + \varepsilon \right]$$

$$+ \mathbb{P}(A > 0, 1 + \varepsilon < r(s, a')) \mathbb{E}_{a' \sim \pi_{old}(\cdot|s), A} \left[ \frac{\partial}{\partial \theta_{s,a}} C_\varepsilon(r(s, a'), A) \mid A > 0, 1 + \varepsilon < r(s, a') \right]$$

$$+ \mathbb{P}(A > 0, 0 \le r(s, a') < 1 - \varepsilon) \mathbb{E}_{a' \sim \pi_{old}(\cdot|s), A} \left[ \frac{\partial}{\partial \theta_{s,a}} C_\varepsilon(r(s, a'), A) \mid A > 0, 0 \le r(s, a') < 1 - \varepsilon \right]$$

$$+ \mathbb{P}(A < 0, 1 - \varepsilon < r(s, a') < 1 + \varepsilon) \mathbb{E}_{a' \sim \pi_{old}(\cdot|s), A} \left[ \frac{\partial}{\partial \theta_{s,a}} C_\varepsilon(r(s, a'), A) \mid A < 0, 0 \le r(s, a') < 1 - \varepsilon \right]$$

$$+ \mathbb{P}(A < 0, 1 + \varepsilon < r(s, a')) \mathbb{E}_{a' \sim \pi_{old}(\cdot|s), A} \left[ \frac{\partial}{\partial \theta_{s,a}} C_\varepsilon(r(s, a'), A) \mid A < 0, 1 + \varepsilon < r(s, a') \right]$$

$$+ \mathbb{P}(A < 0, 0 \le r(s, a) < 1 - \varepsilon) \mathbb{E}_{a' \sim \pi_{old}(\cdot|s), A} \left[ \frac{\partial}{\partial \theta_{s,a}} C_\varepsilon(r(s, a'), A) \mid A < 0, 0 \le r(s, a') < 1 - \varepsilon \right]$$

$$+ \mathbb{P}(A = 0) \mathbb{E}_{a' \sim \pi_{old}(\cdot|s), A} \left[ \frac{\partial}{\partial \theta_{s,a}} C_\varepsilon(r(s, a'), A) \mid A = 0 \right]$$

$$= \mathbb{P}(A > 0, 1 - \varepsilon < r(s, a') < 1 + \varepsilon) \mathbb{E}_{a' \sim \pi_{old}(\cdot|s), A} \left[ \frac{\partial}{\partial \theta_{s,a}} r(s, a') \cdot A \mid A > 0, 1 - \varepsilon < r(s, a') < 1 + \varepsilon \right]$$

$$+ \mathbb{P}(A > 0, 1 + \varepsilon < r(s, a')) \mathbb{E}_{a \sim \pi_{old}(\cdot|s), A} \left[ \frac{\partial}{\partial \theta_{s,a}} (1 + \varepsilon) \cdot A \mid A > 0, 1 + \varepsilon < r(s, a') \right]$$

$$+ \mathbb{P}(A > 0, 0 \le r(s, a') < 1 - \varepsilon) \mathbb{E}_{a' \sim \pi_{old}(\cdot|s), A} \left[ \frac{\partial}{\partial \theta_{s,a}} r(s, a') \cdot A \mid A > 0, 0 < r(s, a') < 1 - \varepsilon \right]$$

$$+ \mathbb{P}(A < 0, 1 - \varepsilon \le r(s, a') < 1 + \varepsilon) \mathbb{E}_{a' \sim \pi_{old}(\cdot|s), A} \left[ \frac{\partial}{\partial \theta_{s,a}} r(s, a') \cdot A \mid A < 0, 1 - \varepsilon \le r(s, a') < 1 + \varepsilon \right]$$

$$+ \mathbb{P}(A < 0, 1 + \varepsilon < r(s, a')) \mathbb{E}_{a' \sim \pi_{old}(\cdot|s), A} \left[ \frac{\partial}{\partial \theta_{s,a}} r(s, a') \cdot A \mid A < 0, 1 + \varepsilon < r(s, a') \right]$$

$$+ \mathbb{P}(A < 0, 0 \le r(s, a') < 1 - \varepsilon) \mathbb{E}_{a' \sim \pi_{old}(\cdot|s), A} \left[ \frac{\partial}{\partial \theta_{s,a}} (1 - \varepsilon) \cdot A \mid A < 0, 0 \le r(s, a') < 1 - \varepsilon \right]$$

Note that $A$ is independent of $\pi_{old}$, and that $\mathbb{E}[A] = 0$. Denote $\mathbb{E}[A|A > 0] = \mu = -\mathbb{E}[A|A < 0]$ and $\mathbb{P}(A > 0) = \mathbb{P}(A < 0) = \nu$. Then the symmetric terms cross out, resulting in

$$
\frac{\partial}{\partial \theta_{s,a}} \mathcal{J}(\theta) = \mathbb{P}(A > 0, 0 \leq r(s,a) < 1 - \varepsilon) \mathbb{E}_{a \sim \pi_{old}(\cdot|s), A} \left[ \frac{\partial}{\partial \theta_{s,a}} r(s,a) \cdot A \mid A > 0, 0 \leq r(s,a) < 1 - \varepsilon \right]
$$

$$
+ \mathbb{P}(A < 0, 1 + \varepsilon < r(s,a)) \mathbb{E}_{a \sim \pi_{old}(\cdot|s), A} \left[ \frac{\partial}{\partial \theta_{s,a}} r(s,a) \cdot A \mid A < 0, 1 + \varepsilon < r(s,a) \right]
$$

$$
= \mu \nu \mathbb{P}(0 \leq r(s,a) < 1 - \varepsilon) \mathbb{E}_{a \sim \pi_{old}(\cdot|s)} \left[ \frac{\partial}{\partial \theta_{s,a}} r(s,a) \mid 0 \leq r(s,a) < 1 - \varepsilon \right]
$$

$$
- \mu \nu \mathbb{P}(1 + \varepsilon < r(s,a)) \mathbb{E}_{a \sim \pi_{old}(\cdot|s)} \left[ \frac{\partial}{\partial \theta_{s,a}} r(s,a) \mid 1 + \varepsilon < r(s,a) \right]
$$

Recall that with $r_k(s,a) = \frac{\pi_k(a|s)}{\pi_{old}(a|s)}$, the probabilistic events corresponding to clipping are denoted as:

$$
X_k(s) = \{a \in \mathcal{A}(s) \mid r_k(s,a) < 1 - \varepsilon_{\text{low}}\}
$$
$$
Y_k(s) = \{a \in \mathcal{A}(s) \mid r_k(s,a) > 1 + \varepsilon_{\text{high}}\}.
$$

Then the above expression simplifies into

$$
\frac{\partial}{\partial \theta_{s,a}} \mathcal{J}(\theta^k) = \mu \nu d^{\pi_{old}}(s) \mathbb{E}_{a' \sim \pi_{old}(\cdot|s)} \left[ \frac{\partial}{\partial \theta_{s,a}} \left( \frac{\pi_k(a'|s)}{\pi_{old}(a'|s)} \right) (\mathbf{1}_{X_k(s)}(a') - \mathbf{1}_{Y_k(s)}(a')) \right]
$$

where $\mathbf{1}_C(x)$ is the indicator function of set $C$. Note that the derivative of $\pi(a|s) = \exp(\theta_{s,a})/\sum_{a' \in \mathcal{A}} \exp(\theta_{s,a'}) = \exp(\theta_{s,a})/Z$ w.r.t. $\theta$ is

$$
\frac{\partial \pi_\theta(a'|s')}{\theta_{s,a}} = \begin{cases} \mathbf{1}_{\{s=s'\}} \cdot \left( \frac{\exp(\theta_{s,a})}{Z} - \frac{\exp(2\theta_{s,a})}{Z^2} \right) & \text{if } a' = a \\ -\mathbf{1}_{\{s=s'\}} \cdot \left( \frac{\exp(\theta_{s,a} + \theta_{s,a'})}{Z^2} \right) & \text{if } a' \neq a \end{cases}
$$

$$
= \mathbf{1}_{\{s=s'\}} \cdot \left( \mathbf{1}_{\{a'=a\}} \frac{\exp(\theta_{s,a})}{Z} - \frac{\exp(\theta_{s,a} + \theta_{s,a'})}{Z^2} \right)
$$

$$
= \mathbf{1}_{\{s=s'\}} \cdot \left( \mathbf{1}_{\{a'=a\}} \pi_\theta(a|s) - \pi_\theta(a|s) \cdot \pi_\theta(a'|s) \right)
$$

$$
= \mathbf{1}_{\{s=s'\}} \pi_\theta(a|s) \left( \mathbf{1}_{\{a'=a\}} - \pi_\theta(a'|s) \right)
$$

Hence,

$$
\frac{\partial}{\partial \theta_{s,a}} \mathcal{J}(\theta^k) = \mu \nu d^{\pi_{old}}(s) \mathbb{E}_{a' \sim \pi_{old}(\cdot|s)} \left[ \left( \mathbf{1}_{\{a=a'\}} \frac{\pi_k(a|s)}{\pi_{old}(a'|s)} - \pi_k(a|s) \frac{\pi_k(a'|s)}{\pi_{old}(a'|s)} \right) (\mathbf{1}_{X_k(s)}(a') - \mathbf{1}_{Y_k(s)}(a')) \right]
$$

$$
= \mu \nu d^{\pi_{old}}(s) \sum_{a' \in \mathcal{A}(s)} \left[ \left( \mathbf{1}_{\{a=a'\}} \pi_k(a|s) - \pi_k(a'|s) \right) (\mathbf{1}_{X_k(s)}(a') - \mathbf{1}_{Y_k(s)}(a')) \right]
$$

$$
= \mu \nu d^{\pi_{old}}(s) \left[ \pi_k(a|s) (\mathbf{1}_{X_k}(a|s) - \mathbf{1}_{Y_k(s)}(a)) - \pi_k(a|s) \mathbb{E}_{a' \sim \pi_k(\cdot|s)} (\mathbf{1}_{X_k(s)}(a') - \mathbf{1}_{Y_k(s)}(a')) \right]
$$

$$
= \mu \nu d^{\pi_{old}}(s) \cdot \pi_k(a|s) \left[ h_k(a|s) - \mathbb{E}_{a' \sim \pi_k(\cdot|s)} h_k(a'|s) \right]
$$

where we define $h_k(a|s) = \mathbf{1}_{X_k(s)}(a) - \mathbf{1}_{Y_k(s)}(a)$.

Recall that as we are assuming gradient descent updates, we update the logits via the policy gradient with respect to the clipped objective

$$
\theta_{s,a}^{k+1} - \theta_{s,a}^k = \eta \cdot \frac{\partial}{\partial \theta_{s,a}} \mathcal{J}(\theta^k),
$$

obtaining the following logit change formula.

$$
\theta_{s,a}^{k+1} - \theta_{s,a}^k = \mu \nu \eta \, d^{\pi_{old}}(s) \, \pi_k(a|s) \left( h_k(a|s) - \mathbb{E}_{a' \sim \pi_k(\cdot|s)} h_k(a'|s) \right)
$$

Now we can plug this this back into equation 9. By direct calculation, we conclude our proof.

$$\mathcal{H}(\theta^{k+1}|s) - \mathcal{H}(\theta^k|s)$$

$$= -\mathbb{E}_{a\sim\pi_k(\cdot|s)}\left[\left(\theta^{k+1}_{s,a} - \theta^k_{s,a}\right)\left(\log\pi_k(a|s) + \mathcal{H}(\theta^k|s)\right)\right] + \mathcal{O}((\Delta\theta)^2)$$

$$= -\mu\nu\eta\ d^{\pi_{old}}(s)\mathbb{E}_{a\sim\pi_k(\cdot|s)}\left[\pi_k(a|s)(h_k(a|s) - \mathbb{E}_{a'\sim\pi_k(\cdot|s)}[h_k(a'|s)])(\log\pi_k(a|s) + \mathcal{H}(\theta^k|s)\right] + \mathcal{O}(\eta^2)$$

$$= -\mu\nu\eta\ d^{\pi_{old}}(s)\Big[\mathbb{E}_{a\sim\pi_k(\cdot|s)}[\pi_k(a|s)\log\pi_k(a|s)h_k(a|s)] + \mathbb{E}_{a\sim\pi_k(\cdot|s)}[\pi_k(a|s)h_k(a|s)]\mathcal{H}(\theta^k|s)$$

$$- \mathbb{E}_{a\sim\pi_k(\cdot|s)}[\pi_k(a|s)\log\pi_k(a|s)]\mathbb{E}_{a\sim\pi_k(\cdot|s)}[h_k(a|s)]$$

$$- \mathbb{E}_{a\sim\pi_k(\cdot|s)}[\pi_k(a|s)]\mathbb{E}_{a\sim\pi_k(\cdot|s)}[h_k(a|s)]\mathcal{H}(\theta^k|s)\Big] + \mathcal{O}(\eta^2)$$

$$= -\mu\nu\eta\ d^{\pi_{old}}(s)\Big[p_k\mathbb{E}_{a\sim\pi_k(\cdot|X_k(s))}[\pi_k(a|s)\log\pi_k(a|s)|X_k(s)] - q_k\mathbb{E}_{a\sim\pi_k(\cdot|Y_k(s))}[\pi_k(a|s)\log\pi_k(a|s)|Y_k(s)]$$

$$+ p_k(s)\mathbb{E}_{a\sim\pi_k(\cdot|X_k(s))}[\pi_k(a|s)|X_k(s)]\mathcal{H}(\theta^k|s) - q_k\mathbb{E}_{a\sim\pi_k(\cdot|Y_k(s))}[\pi_k(a|s)|Y_k(s)]\mathcal{H}(\theta^k|s)$$

$$- p_k(s)(s)(\mathbb{E}_{a\sim\pi_k(\cdot|s)}[\pi_k(a|s)\log\pi_k(a|s)] + \mathbb{E}_{a\sim\pi_k(\cdot|s)}[\pi_k(a|s)]\mathcal{H}(\theta^k|s))$$

$$+ q_k(s)(\mathbb{E}_{a\sim\pi_k(\cdot|s)}[\pi_k(a|s)\log\pi_k(a|s)] + \mathbb{E}_{a\sim\pi_k(\cdot|s)}[\pi_k(a|s)]\mathcal{H}(\theta^k|s))\Big] + \mathcal{O}(\eta^2)$$

$$= \mu\nu\eta\ d^{\pi_{old}}(s)\left(p_k(s)(\mathbb{E}[Q(a,s)] - \mathbb{E}[Q(a,s)|X_k(s)]) - q_k(s)(\mathbb{E}[Q(a,s)] - \mathbb{E}[Q(a,s)|Y_k(s)])\right) + \mathcal{O}(\eta^2)$$

where we define $p_k(s) = \mathbb{P}_{a\sim\pi_k(\cdot|s)}(X_k(s))$, $q_k(s) = \mathbb{P}_{a\sim\pi_k(\cdot|s)}(Y_k(s))$, and $Q(a,s) = \pi_k(a|s)(\log\pi_k(a|s) + \mathcal{H}(\theta^k|s))$. $\qquad\square$

## B  ANALYSIS OF NATURAL POLICY GRADIENT: PROOF OF THEOREM 2

Here we present the proof for Theorem 2

*Proof.* We first obtain the first-order Taylor expansion of policy entropy relative to the policy change $\Delta\pi = \pi_{k+1}(s) - \pi_k(s) := (\pi_{k+1}(a|s) - \pi_k(a|s))_{a\in\mathcal{A}(s)}$. The prior work Cui et al. (2025) has carried out analyses similar to this first step.

$$\mathcal{H}(\pi_{k+1}|s) - \mathcal{H}(\pi_k|s) = \langle\pi_{k+1}(s) - \pi_k(s), \nabla_\pi\mathcal{H}(\pi_k|s)\rangle + \mathcal{O}(\|\Delta\pi\|^2)$$

$$= \sum_{a\in\mathcal{A}(s)}(\pi_{k+1}(a|s) - \pi_k(a|s))\frac{\partial}{\partial\pi_k(a|s)}(-\pi_k(a|s)\log\pi_k(a|s)) + \mathcal{O}(\|\Delta\pi\|^2)$$

$$= -\sum_{a\in\mathcal{A}(s)}(\pi_{k+1}(a|s) - \pi_k(a|s))(\log\pi_k(a|s) + 1) + +\mathcal{O}(\|\Delta\pi\|^2)$$

$$= -\sum_{a\in\mathcal{A}(s)}(\pi_{k+1}(a|s) - \pi_k(a|s))\log\pi_k(a|s) - \sum_{a\in\mathcal{A}(s)}(\pi_{k+1}(a|s) - \pi_k(a|s)) + \mathcal{O}(\|\Delta\pi\|^2)$$

$$= -\mathbb{E}_{a\sim\pi_k(\cdot|s)}\left[\left(\frac{\pi_{k+1}(a|s)}{\pi_k(a|s)} - 1\right)\log\pi_k(a|s)\right] + \mathcal{O}(\|\Delta\pi\|^2) \qquad (10)$$

For our next step (and this is where the technical novelty of our analysis begins), we express the policy ratio $\frac{\pi_{k+1}(a|s)}{\pi_k(a|s)}$ in terms of clipping events. As we are using the natural policy gradient algorithm, the policy is updated as

$$\frac{\pi_{k+1}(a|s)}{\pi_k(a|s)} = \frac{\exp\left(\eta\nabla_{\pi(a|s)}\mathcal{J}(\pi_k)\right)}{\sum_{a'\in\mathcal{A}(s)}\pi_k(a'|s)\exp\left(\eta\nabla_{\pi(a'|s)}\mathcal{J}(\pi_k)\right)}$$

where $\mathcal{J}$ is the clipped surrogate objective

$$\mathcal{J}(\pi) = \mathbb{E}_{x\sim\mathcal{D}, y\sim\pi_{old}(\cdot|x), A}\left[\frac{1}{T}\sum_{t=0}^{T}C_\varepsilon(r_t, A_t)\right]$$

with $r_t = \frac{\pi(y_t|y_{<t},x)}{\pi_{old}(y_t|y_{<t},x)}$. Now we can simplify this as

$$\frac{\partial}{\partial\pi(a|s)}\mathcal{J}(\pi) = \mathbb{E}_{x\sim\mathcal{D},\tau\sim\pi_{old}(\cdot|x),A}\left[\frac{1}{T}\frac{\partial}{\partial\pi(a|s)}\sum_{t=0}^{T}C_\varepsilon(r_t,A_t)\right]$$

$$= \mathbb{E}_{x\sim\mathcal{D},\tau\sim\pi_{old}(\cdot|x),A}\left[\frac{1}{T}\sum_{t=0}^{T}\frac{\partial}{\partial\pi(a|s)}\mathbf{1}_{\{(y_{<t},x)=s\}}\mathbf{1}_{\{y_t=a\}}C_\varepsilon(r_t,A_t)\right]$$

$$= \mathbb{E}_{x\sim\mathcal{D},y_t\sim\pi_{old}(\cdot|y_{<t},x),A_t}\left[\mathbf{1}_{\{(y_{<t},x)=s\}}\frac{\partial}{\partial\pi(a|s)}C_\varepsilon(r_t,A_t)\right]$$

$$= d^{\pi_{old}}(s) \times \mathbb{E}_{a'\sim\pi_{old}(\cdot|s),A}\left[\mathbf{1}_{\{a'=a\}}\frac{\partial}{\partial\pi(a|s)}C_\varepsilon(r(s,a'),A)\right]$$

$$= d^{\pi_{old}}(s)\pi_{old}(a|s) \times \mathbb{E}_A\left[\frac{\partial}{\partial\pi(a|s)}C_\varepsilon(r(s,a),A)\right]$$

where $d^{\pi_{old}}(s)$ is the state-visiting probability under the policy $\pi_{old}$. Now expanding $C_\varepsilon(r,A)$ as

$$C_\varepsilon(r,A) = \mathbf{1}_{A\geq 0}\cdot A\cdot(r\cdot\mathbf{1}_{r\leq 1+\varepsilon}+(1+\varepsilon)\cdot\mathbf{1}_{r>1+\varepsilon})+\mathbf{1}_{A<0}\cdot A\cdot(r\cdot\mathbf{1}_{r\geq 1-\varepsilon}+(1-\varepsilon)\cdot\mathbf{1}_{r<1-\varepsilon})$$

we have

$$\mathbb{E}_A\left[\frac{\partial}{\partial\pi(a|s)}C_\varepsilon(r(s,a),A)\right] = \mathbb{E}_A\left[\mathbf{1}_{A\geq 0}\cdot A\left(\mathbf{1}_{r<1+\varepsilon}\cdot\frac{\partial r}{\partial\pi(a|s)}\right)+\mathbf{1}_{A<0}\cdot A\left(\mathbf{1}_{r>1-\varepsilon}\cdot\frac{\partial r}{\partial\pi(a|s)}\right)\right]$$

$$= \mathbb{P}(A\geq 0)\cdot\mathbb{E}_A\left[\left(\frac{\mathbf{1}_{r<1+\varepsilon}}{\pi_{old}(a|s)}\right)\cdot A\,\Big|\,A\geq 0\right]+\mathbb{P}(A<0)\cdot\mathbb{E}_A\left[\left(\frac{\mathbf{1}_{r>1-\varepsilon}}{\pi_{old}(a|s)}\right)\cdot A\,\Big|\,A<0\right]$$

$$= \frac{\mu\nu}{\pi_{old}(a|s)}\{(1-\mathbf{1}_{Y(s)}(a))-(1-\mathbf{1}_{X(s)}(a))\}$$

$$= \frac{\mu\nu}{\pi_{old}(a|s)}(\mathbf{1}_{X(s)}(a)-\mathbf{1}_{Y(s)}(a))$$

Therefore we have

$$\frac{\partial}{\partial\pi(a|s)}\mathcal{J}(\theta) = \mu\nu d^{\pi_{old}}(s)(\mathbf{1}_{X_k(s)}(a)-\mathbf{1}_{Y_k(s)}(a))$$

and therefore the logit change can be written as

$$\frac{\pi_{k+1}(a|s)}{\pi_k(a|s)} = \frac{e^{\mu\nu\eta d^{\pi_{old}}(s)(\mathbf{1}_{X_k(s)}(a)-\mathbf{1}_{Y_k(s)}(a))}}{\sum_{a\in\mathcal{A}(s)}\pi_k(a|s)e^{\mu\nu\eta d^{\pi_{old}}(s)(\mathbf{1}_{X_k(s)}(a)-\mathbf{1}_{Y_k(s)}(a))}}$$

Now we can plug this this back into equation 9.

$$\mathcal{H}(\pi_{k+1}|s)-\mathcal{H}(\pi_k|s) = -\mathbb{E}_{a\sim\pi_k(\cdot|s)}\left[\left(\frac{\pi_{k+1}(a|s)}{\pi_k(a|s)}-1\right)\log\pi_k(a|s)\right]+\mathcal{O}(\|\Delta\pi\|^2)$$

$$= -\mathbb{E}_{a\sim\pi_k(\cdot|s)}\left[\left(\frac{e^{\mu\nu\eta d^{\pi_{old}}(s)(\mathbf{1}_{X_k(s)}(a)-\mathbf{1}_{Y_k(s)}(a))}}{\sum_{a\in\mathcal{A}(s)}\pi_k(a|s)e^{\mu\nu\eta d^{\pi_{old}}(s)(\mathbf{1}_{X_k(s)}(a)-\mathbf{1}_{Y_k(s)}(a))}}-1\right)\log\pi_k(a|s)\right]+\mathcal{O}(\|\Delta\pi\|^2)$$

Here notice that

$$\sum_{a\in\mathcal{A}(s)}\pi_k(a|s)e^{\mu\nu\eta d^{\pi_{old}}(s)(\mathbf{1}_{X_k(s)}(a)-\mathbf{1}_{Y_k(s)}(a))} = \underbrace{e^{\mu\nu\eta d^{\pi_{old}}(s)}\mathbb{P}(X_k)+e^{-\mu\nu\eta d^{\pi_{old}}(s)}\mathbb{P}(Y_k)+(1-\mathbb{P}(X_k)-\mathbb{P}(Y_k))}_{:=Z^k(s)}$$

, in other words this is a quantity determined soley by the portion of actions under $s$ that clip-highed and clip-lowed. Thus denoting this value as $Z^k(s)$, we can simplify this equation as:

$$\mathcal{H}(\pi_{k+1}|s) - \mathcal{H}(\pi_k|s) \approx - \Bigg( \frac{e^{\mu\nu\eta d^{\pi_{old}}(s)} - 1}{Z^k(s)} \mathbb{E}_{a\sim\pi_k(\cdot|s)}[\log\pi_k(a|s)|X_k]\mathbb{P}(X_k)$$

$$- \frac{1 - e^{-\mu\nu\eta d^{\pi_{old}}(s)}}{Z^k(s)} \mathbb{E}_{a\sim\pi_k(\cdot|s)}[\log\pi_k(a|s)|Y_k]\mathbb{P}(Y_k) + \left(1 - \frac{1}{Z^k(s)}\right)\mathcal{H}(s) \Bigg)$$

Now applying again the second order approximation $e^{\mu\nu\eta d^{\pi_{old}}(s)} - 1 \approx \mu\nu\eta d^{\pi_{old}}(s)$, $e^{-\mu\nu\eta d^{\pi_{old}}(s)} - 1 \approx -\mu\nu\eta d^{\pi_{old}}(s)$ , we can simplify this relation to

$$\mathcal{H}(\pi^{k+1}|s) - \mathcal{H}(\pi^k|s) \approx -\delta \left( \mathbb{P}(X_k)(\mathbb{E}_{a\sim\pi_k(\cdot|s)}[\log\pi_k|X_k] + \mathcal{H}(\pi^k|s)) - \mathbb{P}(Y_k)(\mathbb{E}_{a\sim\pi_k(\cdot|s)}[\log\pi_k|Y_k] + \mathcal{H}(\pi^k|s)) \right)$$

where $\approx$ represents first order approximation over $\eta$, and $\delta = \mu\nu\eta d^{\pi_{old}}(s)$. $\qquad\square$

## C  Experimental Settings and Additional Experimental Results

### C.1  Experimental setup

For the random reward RL training experiments, we used the `GSM8K` dataset as the traning dataset, and conducted experiments with base models `Qwen2.5-1.5B-Instruct` (Yang et al., 2024) and `Llama-3.2-1B-Instruct` (Grattafiori et al., 2024). For general mathematical reasoning tasks, we train the `Qwen2.5-7B-Instruct` model with the `DAPO-Math-17k` (Yu et al., 2025) dataset, and validate it on `MATH-500` (Hendrycks et al., 2021), `AMC23` (AI-MO), `AIME2024`, and `AIME2025` datasets (HuggingFaceH4). We also train `Qwen2.5-3B-Instruct` and `Llama-3-8B-Instruct` model with the `GSM8K` dataset, and validate it on the `GSM8K` (Cobbe et al., 2021) test dataset. For validation, we perform string match for the last numerical value for `GSM8K` test datasets, and use the `Math-Verify` (HuggingFace, 2025) package.

We use different training configurations for the `GSM8K` and `DAPO-MATH-17k` dataset, and separate them with /. In Table 1, we provide the training and generation details for the experiments in the paper. For all experiments, KL divergence loss or entropy regularization loss were not deployed.

| Hyperparameter | Value |
|---|---|
| Optimizer | AdamW |
| Learning rate | $5 \times 10^{-7}$ / $1 \times 10^{-6}$ |
| GRPO batch size | 512 |
| Optimizer batch size | 256 |
| Policy updates per rollout | 16 |
| Group Size | 8 |
| Max response length | 4096 |
| Temperature (train) | 1.0 |
| Temperature (validation) | 1.0 |
| Top p (train) | 1.0 |
| Top p (validation) | 0.95 |
| Dynamic Sampling | None / True |
| Overlong penalty factor | None / 1.0 |

Table 1: Training configurations used for `GSM8K` dataset / `DAPO-Math-17k` dataset.

### C.2  Random reward training across different settings

To corroborate that the entropy minimization effect of random rewards with symmetric clipping $\varepsilon_{\text{low}} = \varepsilon_{\text{high}}$ is not a model-agnostic result, we conduct the same experiment with three base models from different model families. In the left panel of Figure 4, we present the normalized entropy of models `Qwen2.5-1.5B-Instruct`, `Llama3.2-1B-Instruct`, and

`OLMo-2-0425-1B-Instruct` during RL training. We normalize the entropy of each model by the entropy of the base model. Due to slow convergence, we set $\varepsilon_{\text{high}} = \varepsilon_{\text{low}} = 0.1$ for `Olmo2`, and $\varepsilon_{\text{high}} = \varepsilon_{\text{low}} = 0.2$ for other models. One can clearly observe a decreasing trend for all three models.

Further, we use different random sources for the rewards for RL training of `Qwen2.5-1.5B-Instruct` model. We test three random sources from which we sample the rewards: Bernoulli random reward with $p = 0.3$ ('Bernoulli $p = 0.3$) and $p = 0.7$ ('Bernoulli $p = 0.7$) where reward 1 is given for probability $p$ and 0 for probability $1 - p$, and standard normal distribution ('Gaussian') so that $r \sim \mathcal{N}(0, 1)$. As in other experiments, we use the GRPO algorithm with group size 8. In the right panel of Figure 4, one can conclude that entropy minimization is implicitly performed during the RL training, regardless of the distribution from which the reward is sampled.

## C.3    ADDITIONAL EXPERIMENTS FOR LLAMA BASE MODELS

In this section, we provide further experimental results that validate our findings. Specifically, we reproduce the main figures in the paper with `Llama` base models. In Figure 8 (a), we conduct random reward experiments with base model `Llama3.2-1B-Instruct`. As in the case with `Qwen`-based models, we can clearly observe the opposite effects of upper and lower clip on policy entropy. Figure 8 (b) shows results for the same experiments for nonrandom rewards, trained on the GSM8K dataset with the `Llama3-8B-Instruct` model. This is associated with the result in Figure 8 (c) where the performance and pass rate estimated with $k = 8$ for different clipping settings are presented. We can again observe that RLVR proceeds and decreases entropy the `pass@k` rate typically decreases. However, by aggressively utilizing clip-low configurations, the rate of `pass@k` rate decreasing could be greatly mitigated without compromising the average reasoning ability.

## C.4    ADDITIONAL EXPERIMENTS FOR DAPO-MATH-17K TRAINING DATASET

Here, we present additional experimental results for `Qwen2.5-7B-Instruct` model trained with the `DAPO-Math-17k` dataset. In Figure 9, we present the result of the clipping ablation experiment observing the entropy dynamics. As expected, we can clearly observe the clipping bias on entropy. In Figure 10, we further provide the validation results for the `mean@32` and `pass@32` metric. Similar to other validation benchmark, deliberate clipping for increased policy entropy effectively hinders exploration degradation throughout the training.

(a)

(b)

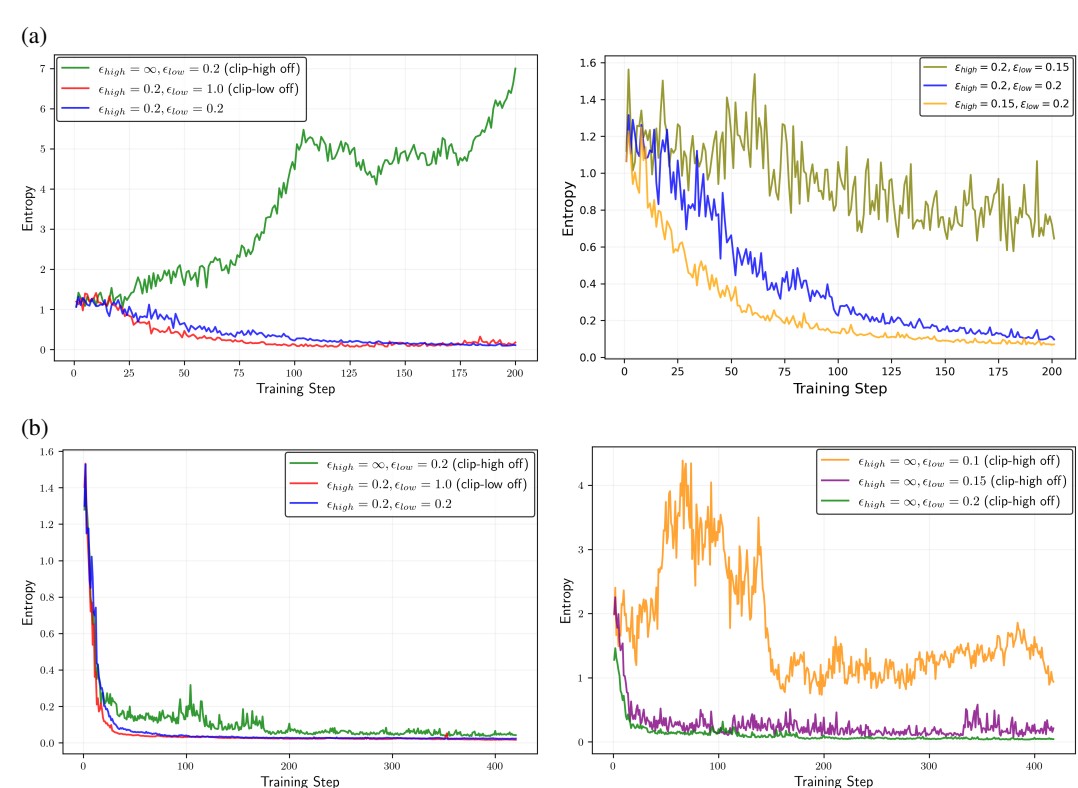

Figure 8: Main experimental results with `Llama` base models. Policy entropy change during RL training with **(a)** random rewards for `Llama3.2-1B-Instruct` and **(b)** general RLVR rewards for `Llama3-8B-Instruct` model. For both random and nonrandom rewards, we observe a clear trend of clip-low increasing entropy and clip-high decreasing it.

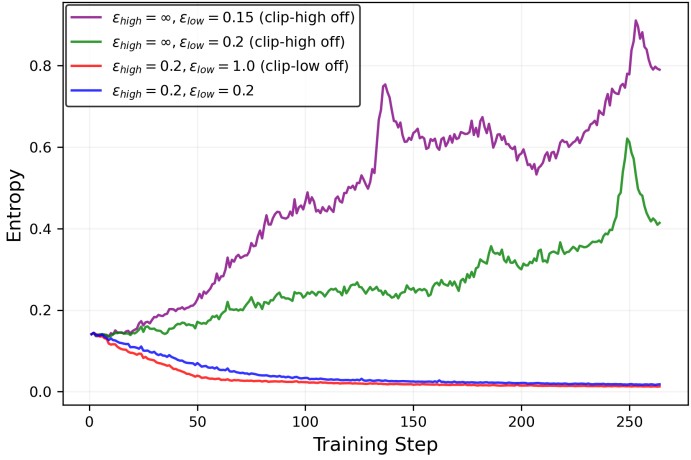

Figure 9: Clip ablation study for the entropy dynamics of `Qwen2.5-7B-Instruct` trained on the `DAPO-Math-17k` dataset.

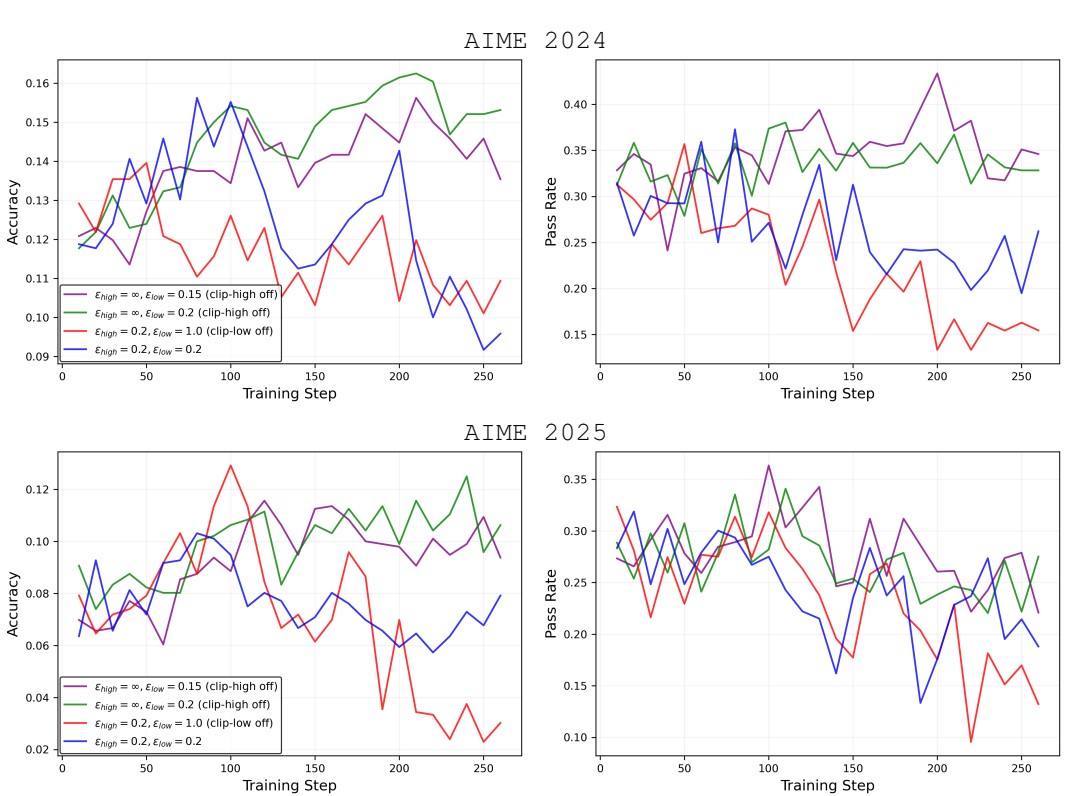

Figure 10: Performance measured by the `mean@8` metric **(left)** and `pass@8` metric **(right)** metric during RLVR for the `Qwen2.5-7B-Instruct` model trained with `DAPO-Math-17k` dataset, evaluated on the `AIME 2024` and `AIME 2025` datasets.