# OpenReview forum: "Clip-Low Increases Entropy and Clip-High Decreases Entropy in  Reinforcement Learning of Large Language Models"
_ICLR.cc/2026/Conference — ICLR 2026 Conference Withdrawn Submission_

### Official Review · Reviewer_88eX · 2025-10-27

**Soundness:** 2
**Presentation:** 1
**Contribution:** 2
**Rating:** 2
**Confidence:** 4

**Summary:**

This paper investigates the phenomenon of entropy collapse during RLVR and other PPO/GRPO-based fine-tuning methods for LLMs. hey argue that: clip-low events increase entropy, clip-high events decrease entropy. It develops a theoretical analysis using first-order approximations of the entropy change under both policy gradient and natural policy gradient formulations. And it further conducts experiments under random and real reward settings, showing that clip-high effects dominate in practice, leading to entropy decay.

**Strengths:**

1. Timely and important research problem. Entropy collapse is a well-recognized but insufficiently studied bottleneck in RL for LLMs. Analyzing its source from the perspective of clipping bias is novel and relevant to current alignment trends.

2. Conceptual and theoretical novelty. The analytical approach separating the entropic effects of clip-low vs. clip-high is fresh and offers new intuition into PPO/GRPO dynamics that are often treated as black boxes.

**Weaknesses:**

1. Poor presentation and structure. The introduction and abstract are long-winded and unfocused—readers must wade through pages to grasp the core message. Variable notation is inconsistent and densely packed, making formulas difficult to follow.

2. Theoretical development lacks clarity. Theorems 1 and 2 depend on several assumptions that are only heuristically justified. The random-reward analysis, while interesting, is quite distant from realistic RLVR conditions; actually missing a theoretical bridge between the two regimes.

3. Weak data visualization and empirical reporting. Many results are described qualitatively rather than quantitatively, with no statistical significance reported. Experiments look like exploratory plots rather than rigorous validation.

4. Poor writing quality and technical formatting. In particular, the proof in Pages 13-18 exceeds the page limit in many many places.

**Questions:**

Please refer to the **Weaknesses** part.
Overall, I believe this paper is novel and meaningful, but the presentation is really really poor!!! The authors are encouraged to improve the presentation of the paper. I am also looking forward to the authors’ response, and may reconsider this paper based on it.

---

### Official Review · Reviewer_fY5M · 2025-11-01

**Soundness:** 4
**Presentation:** 3
**Contribution:** 2
**Rating:** 4
**Confidence:** 4

**Summary:**

This paper investigates the phenomenon of entropy collapse in large language models trained with Reinforcement Learning with Verifiable Rewards (RLVR). The authors posit that the PPO/GRPO clipping mechanism is a primary, overlooked driver of this behavior. Through a theoretical analysis in a simplified random-reward setting, the paper demonstrates that "clip-low" (for negative advantages) induces an entropy-increasing bias, while "clip-high" (for positive advantages) induces an entropy-decreasing one. The authors find that the clip-high effect typically dominates, causing entropy to drop.

They empirically validate these findings on mathematical reasoning tasks, showing that tuning the clipping parameters (e.g., disabling clip-high and using a more aggressive clip-low) can stabilize entropy. This control is shown to prevent the degradation of the pass@k metric, thereby maintaining exploration without harming mean@k performance.

**Strengths:**

1. The paper identifies clear biases of clip-low and clip-high to explain the issue of entropy collapse in RLVR. The core idea is simple and easy to follow.
2. The authors successfully validate the proposed findings in a practical, real-world RLVR setting (i.e., mathematical reasoning).

**Weaknesses:**

1. The finding that clip-high decreases entropy is not new. Many prior works (e.g., DAPO) have already identified this mechanism as a driver of entropy decay and proposed mitigations already (e.g., clip-higher).
2. The theoretical analysis, which spans half of the paper, is based on purely random rewards. This is a significant oversimplification, as it completely decouples the reward signal from the policy.
3. The experiments show that this method preserves exploration, but they do not convincingly demonstrate that this leads to a superior final model or a SOTA performance in reasoning performance.

Overall, I agree that the findings are systematic and useful, but silghly doubt if it meets the bar of this venue.

**Questions:**

See above weaknesses.

---

### Official Review · Reviewer_RvR7 · 2025-11-01

**Soundness:** 2
**Presentation:** 1
**Contribution:** 2
**Rating:** 2
**Confidence:** 4

**Summary:**

This paper provides a mechanistic understanding of entropy collapse in RLVR training by theoretically and empirically demonstrating that the 'clip-low' mechanism increases entropy while 'clip-high' decreases it

**Strengths:**

The paper's theoretical contributions are notable—specifically, showing in Theorem 1 and Theorem 2 that both policy gradient and natural policy gradient adhere to the 'Clip-Low Increases Entropy, Clip-High Decreases Entropy' principle.

**Weaknesses:**

- The draft appears careless. Many equations lack proper numbering, and the notation is often inconsistent and confusing. This makes the theoretical arguments difficult to follow.
- Beyond the theoretical analysis, the paper lacks novel insights. The core idea that asymmetric clipping (i.e., 'clip-higher') can enhance exploration is not a new contribution; it was previously introduced by DAPO (Yu et al., 2025). Given that the paper's main contribution appears to be a theoretical validation of a known empirical finding, I am not convinced that the overall contribution is sufficient to meet the acceptance bar.

**Questions:**

See weakness

---

### Official Review · Reviewer_dAE3 · 2025-11-03

**Soundness:** 2
**Presentation:** 2
**Contribution:** 2
**Rating:** 2
**Confidence:** 3

**Summary:**

This paper investigates the phenomenon of entropy collapse in reinforcement learning fine-tuning of large language models (LLMs) using verifiable rewards (RLVR). The core finding is that the clipping mechanism in PPO/GRPO algorithms intrinsically biases the policy’s entropy: the lower clipping threshold (clip-low for negative advantages) tends to increase entropy, while the upper clipping threshold (clip-high for positive advantages) tends to decrease entropy. Through a theoretical analysis in a toy setting with random rewards, the authors prove that clipping alone can drive these entropy changes even when the reward signal carries no information. They then empirically confirm that this effect extends to practical RLVR training on mathematical reasoning tasks.

**Strengths:**

1. The authors derive analytical results in a simplified scenario (random rewards) demonstrating that clip-low and clip-high have opposite effects on policy entropy.
2. The writing is generally clear and well-organized.

**Weaknesses:**

1. The paper focuses on very recent works in 2024–2025 about RLVR and entropy (e.g. DAPO, ProRL, etc.), but it does not mention earlier or broader related research.
2. Because key related works were omitted, readers might not fully grasp how this approach compares with adding an entropy bonus or KL penalty – a short clarification on these points would improve understanding.

**Questions:**

1. The theoretical analysis assumes random, symmetric rewards and a tabular softmax policy. How robust are the conclusions when these assumptions are relaxed? For example, if the reward is not symmetric or if the model is large and finetuned (non-tabular), do the qualitative effects (clip-low raising entropy, clip-high lowering it) still hold reliably?
2. The paper proposes adjusting clipping parameters as a way to control entropy. Have the authors compared this approach to the more standard method of explicit entropy regularization or KL-control during training?
3. The experiments focus on math problem solving. Would the proposed entropy control work for other domains (e.g. code generation or general QA) where exploration is also important? The paper would benefit from a discussion of any limitations – for example, cases where maintaining high entropy might negatively impact performance or lead to undesired behavior. Can the authors comment on potential trade-offs or failure modes.

---

### Note · Authors · 2025-11-19

**Comment:**

We thank the reviewers for their helpful comments and suggestions. We will incorporate this feedback to strengthen the paper.

**Withdrawal Confirmation:**

I have read and agree with the venue's withdrawal policy on behalf of myself and my co-authors.